# Single-molecule imaging and quantification of the immune-variant adhesin VAR2CSA on knobs of *Plasmodium falciparum*-infected erythrocytes

Cecilia P. Sanchez[1], Christos Karathanasis[2], Rodrigo Sanchez[1], Marek Cyrklaff[1], Julia Jäger[3], Bernd Buchholz[4], Ulrich S. Schwarz [3,5], Mike Heilemann [2,5] & Michael Lanzer [1]

PfEMP1 (erythrocyte membrane protein 1) adhesins play a pivotal role in the pathophysiology of falciparum malaria, by mediating sequestration of *Plasmodium falciparum*-infected erythrocytes in the microvasculature. PfEMP1 variants are expressed by *var* genes and are presented on membrane elevations, termed knobs. However, the organization of PfEMP1 on knobs is largely unclear. Here, we use super-resolution microscopy and genetically altered parasites expressing a modified *var2csa* gene in which the coding sequence of the photo-activatable mEOS2 was inserted to determine the number and distribution of PfEMP1 on single knobs. The data were verified by quantitative fluorescence-activated cell sorting analysis and immuno-electron microscopy together with stereology methods. We show that knobs contain $3.3 \pm 1.7$ and $4.3 \pm 2.5$ PfEMP1 molecules, predominantly placed on the knob tip, in parasitized erythrocytes containing wild type and sickle haemoglobin, respectively. The ramifications of our findings for cytoadhesion and immune evasion are discussed.

[1] Center of Infectious Diseases, Parasitology, Universitätsklinikum Heidelberg, Im Neuenheimer Feld 324, 69120 Heidelberg, Germany. [2] Institute for Physical and Theoretical Chemistry, Goethe-University Frankfurt, Max-von-Laue-Str. 7, 60438 Frankfurt, Germany. [3] Institute for Theoretical Physics, Heidelberg University, Philosophenweg 16, 69120 Heidelberg, Germany. [4] Department of Hematology and Oncology, University Children's Hospital, Medical Faculty Mannheim, Theodor-Kutzer-Ufer 1-3, 68167 Mannheim, Germany. [5] BioQuant-Center for Quantitative Biology, Heidelberg University, Im Neuenheimer Feld 267, 69120 Heidelberg, Germany. Correspondence and requests for materials should be addressed to M.H. (email: heilemann@chemie.uni-frankfurt.de) or to M.L. (email: michael.lanzer@med.uni-heidelberg.de)

The virulence of the human malaria parasite *Plasmodium falciparum* is associated with altered rheological properties of infected erythrocytes[1]. Whereas uninfected red blood cells circulate through the vascular system, erythrocytes infected with *P. falciparum* develop cytoadhesive properties ~16–20 h post invasion[2,3] and sequester in the microvasculature to avoid passage through, and clearance by, the spleen. Cytoadhering erythrocytes can obstruct the blood flow and cause impaired tissue perfusion, which eventually can lead to cerebral malaria and other life-threatening complications[4].

Cytoadhesion of parasitized erythrocytes is mediated by immunovariant surface proteins of which the *var* genes-encoded PfEMP1 (erythrocyte membrane protein 1) adhesins constitute the most-prominent adhesin family[1]. Members of the PfEMP1 family can interact with receptors on the surface of microvascular endothelial cells and uninfected erythrocytes, including ICAM-1, CD36, EPCR, and CR1[1,5]. This functional specialization is facilitated by the PfEMP1 domain structure, with defined domains mediating distinct cytoadhesion phenotypes[1,5]. PfEMP1 is presented on the host cell surface in membrane protrusions, termed knobs, which anchor PfEMP1 to the membrane skeleton of the host erythrocyte for mechanical support under flow[6,7]. PfEMP1 variants are considered targets of de-adhesion drugs and anti-disease vaccines[8–10]. For instance, a vaccine against VAR2CSA, the PfEMP1 variant expressed predominantly in parasitized erythrocytes sequestering in the intervillous space of the placenta[11,12], is in clinical development to protect women and their unborn children from maternal malaria[13].

In spite of the medical relevance, very little is known about the organization of PfEMP1 on knobs. Neither the absolute number of PfEMP1 molecules per knob nor their spatial arrangement has been established. Estimates of how many PfEMP1 molecules might be placed on a single knob range from half a dozen to more than 100 copies[14–16]. It is further unclear how the enlarged knobs, as are found in parasitized erythrocytes containing haemoglobin S or C[17–21], affect the number and distribution of PfEMP1 molecules. Enlarged and widely dispersed knobs are associated with a reduced capacity of parasitized haemoglobinopathic erythrocytes to engage in cytoadhesive interactions, and are thought to contribute to the malaria-protective function of sickle cell haemoglobin and related haemoglobinopathies[17,18,21,22]. This dearth of information has been partly owing to a shortage of enabling technology to visualize single PfEMP1 molecules. Recent advances in super-resolution microscopy and image analysis now provide the tools to count single molecules and determine their spatial arrangement[23].

Here, we have used photoactivated light microscopy (PALM) of genetically engineered parasites expressing a modified *var2csa* gene in which the coding sequence of two copies of mEOS2 was inserted to study the numerical and spatial organization of PfEMP1 molecules on single knobs in infected erythrocyte containing wild type or sickle cell haemoglobin. Our data show that knobs contain an unexpected low number of VAR2CSA molecules (3.3 ± 1.7 and 4.3 ± 2.5 molecules per knob in parasitized erythrocytes containing wild type and sickle haemoglobin, respectively) that are predominantly placed at the tip of the knob. Mathematical simulations suggest that this spatial organization has evolved to evade antibody-mediated immune effector mechanisms and increase the capture radius of single VAR2CSA molecules and, thus, receptor binding.

## Results

### Super-resolution imaging of VAR2CSA on single knobs. We
initially explored four different regions within the *var2csa* gene of the *P. falciparum* FCR3 strain for opportunities to accept two

consecutive copies of the coding sequence of the photo-activatable fluorescence protein mEOS2 (for improved signal to noise ratio), without affecting expression and trafficking of the resulting tagged VAR2CSA protein. The mEOS2 tag was introduced by homologous recombination using CRISPR/Cas9-mediated genome-editing technology and site-specific guide RNAs (supplementary Table 1). Attempts to insert the tag into the DBL6ε domain (after amino acid 2634, supplementary Fig. 1) failed, resulting in a chromosome truncation event in which the end of chromosome 12 including most of the ectodomain-encoding sequence of *var2csa* was deleted. In comparison, the tag could be successfully inserted into the region between the DBL5ε and DBL6ε domains (following amino acid 2588), the DBL5ε domain (following amino acid 2310), and immediately after the N-terminal segment (NTS, following amino acid 59) (Fig. 1 and supplementary Fig. 1). Three independent clonal lines were obtained in each case and the respective insertion events were confirmed by sequence analysis of the mutated *var2csa* gene. However, only the clones that contained the mEOS2 tag inserted between the NTS and the DBL1x domain were suitable for further analysis (Fig. 1a). The other two insertion mutants did not present detectable levels of VAR2CSA on the surface.

In the case of the clonal NTS-mEOS2, mutant selected for further analysis, henceforth termed G6, RT-PCR of isolated mRNA confirmed transcription of the genetically altered *var2csa* gene (Fig. 1b and supplementary Fig. 2) and immuno-fluorescence assays of live-infected erythrocytes demonstrated surface labeling, using a monoclonal antibody specific to the DBL3-X domain of VAR2CSA[24] (Fig. 1c). Immuno-electron microscopy, using the same monoclonal antibody, provided further evidence of surface-presented, knob-associated VAR2CSA (Fig. 1d). The knobs displayed by G6 were comparable in terms of diameter and density with those found in erythrocytes infected with the parental FCR3 strain, as shown by scanning electron microscopy (SEM) (supplementary Fig. 3a and b).

Figure 2 shows bright-field and PALM images of representative infected erythrocyte at the trophozoite stage (24–30 h post invasion). PALM images were reconstructed from 12,000 imaging frames recorded with a frame rate of 10 Hz. Owing to the total internal reflection fluorescence (TIRF) configuration, fluorescence signals were predominantly recorded at the plane of the cell surface with a light penetration depth of ~ 100 nm. Knobs, which were 26 ± 13 nm and 46 ± 20 nm in height in G6-infected HbAA and HbAS erythrocytes (supplementary Fig. 3c), were therefore within the TIRF field of view.

Numerous fluorescence clusters, scattered over the entire surface of the parasitized erythrocyte, were observed for HbAA and HbAS erythrocytes infected with G6 (Fig. 2a, b). In contrast, no specific PALM fluorescence signals were seen for erythrocytes infected with age-matched wild-type FCR3 (Fig. 2c). The fluorescence clusters observed for G6 had a mean diameter of 54 ± 27 nm and 64 ± 31 nm in HbAA- and HbAS-infected erythrocytes, respectively[25] (Fig. 2d; $p < 0.001$ according to the Mann–Whitney rank sum test). These values are smaller than the corresponding average knob diameters of 80 ± 12 nm and 104 ± 16 nm, respectively, as determined by SEM (supplementary Fig. 3b), and might suggest a central position of PfEMP1 molecules on knobs (supplementary Fig. 4). A localization of PfEMP1 molecules along the base of the knobs could be excluded on the basis of the spatial resolution of ~ 30 nm of the PALM images (estimated from an average experimental localization precision of 13.1 nm[26]) and the size of the fluorescence clusters that were smaller than the average size of knobs (supplementary Fig. 4).

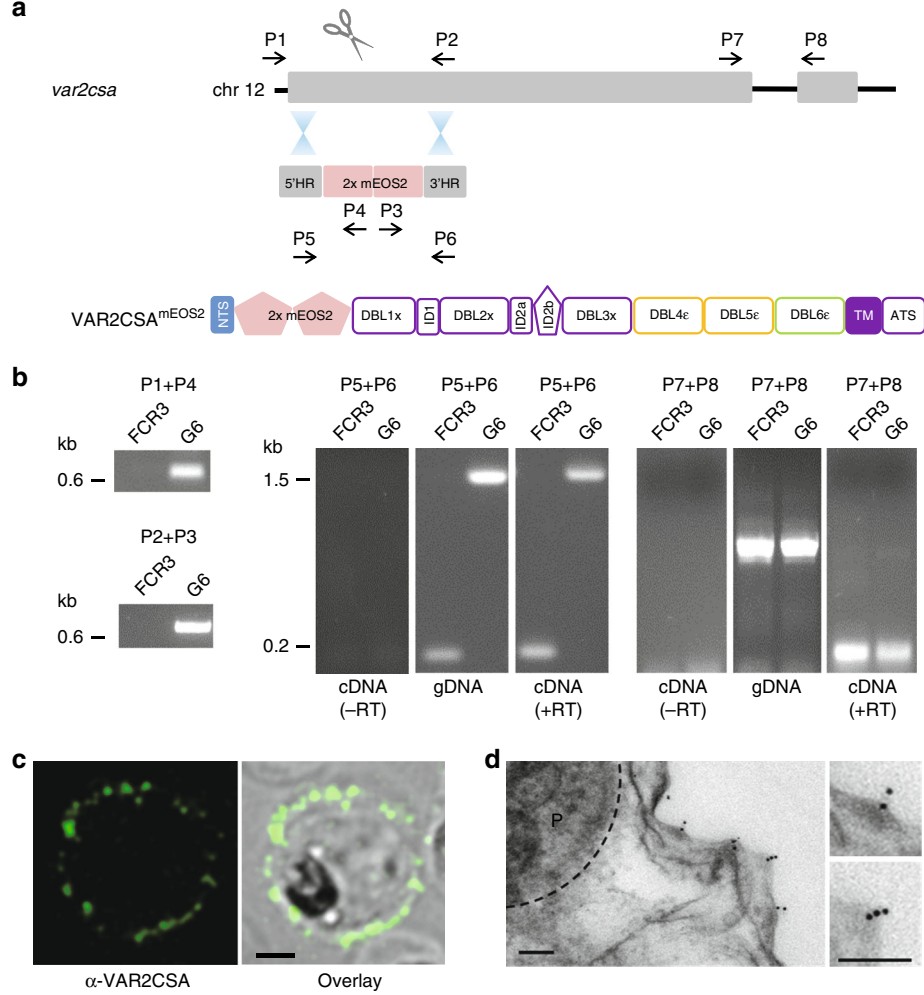

**Fig. 1** Generation of a *P. falciparum* mutant expressing a genomically encoded VAR2CSA-mEOS2 fusion protein. **a** The transfection strategy, using CRISPR/Cas9 genome editing technology, is outlined to insert two copies of the mEOS2 coding sequence between the regions encoding the NTS and the DBL1x domain of *var2csa*. The domain structure of the resulting VAR2CSA$^{mEOS2}$ is shown. The positions of relevant primers for analysis are indicated. **b** The integration event was verified by PCR, using genomic DNA or total mRNA from the resulting mutant, termed G6, and the parental FCR3 strain. The primer pairs used are indicated. Where indicated, RT-PCR was performed in the presence and absence of reverse transcriptase (RT). A size marker is indicated in kilo base pairs (kb). **c** Representative confocal immuno-florescence image of a live HbAA erythrocyte infected with G6 stained with the Zenon-labeled monoclonal antibody PAM 8.1 that is specific to the DBL3-X domain of VAR2CSA[24]. A mid-sectional plane is shown. Bar, 2 µm. **d** A representative immuno-electron micrograph of G6 showing labelling of knobs with the anti-VAR2CSA monoclonal antibody followed by a goat anti human antibody coupled to 10 nm protein A gold. Bar, 100 nm

**Knobs contain only a few VAR2CSA molecules**. Given that each PfEMP1 molecule will occupy a certain space on a knob, the single-molecule detection event is expected to correlate with the fluorescence cluster size, as is indeed observed for both parasitized HbAA and HbAS erythrocytes ($r^2 = 0.74$) (Fig. 3a). To convert the PALM signals into molecular numbers, it is necessary to account for over- and undercounting of mEOS2 signals owing to on–off blinking and incomplete detection. We determined the average number of detection events ("blinking") from single mEOS2 molecules attached to a poly-lysine surface (Fig. 3b), which yielded that on average a single mEOS2 molecule is detected 2.5 times under our imaging conditions. Previous studies have shown that the number of blinking events of mEOS2 molecules attached to poly-lysine surfaces is in very good agreement with values determined in cells using various reference proteins, such that this parameter can be used for molecular quantification of cellular proteins[27,28]. We further considered that (i) the detection efficiency of mEOS2 is in the range of 70%[27–29],

and that (ii) two mEOS2 molecules are attached to each PfEMP1 molecule.

Taking these corrections into consideration, and explicitly correcting for the average number of 2.5 detection events per mEOS2 fluorophore, we obtained on average 3.3 ± 1.7 PfEMP1 molecules per knob for parasitized HbAA erythrocytes (Fig. 3c). This value is based on the measurement of 771 fluorescence clusters from 130 cells, recorded in three sessions, each using blood from different donors for infection with the G6 mutant strain (the number of knobs and cells investigated per session were as follows: 225, 32; 323, 48; and 223, 50). In comparison, HbAS erythrocytes infected with G6 displayed on average 4.3 ± 2.5 PfEMP1 molecules per knob (Fig. 3c) (number of knobs and cells investigated per session: 170, 26; and 468, 50). This value is significantly different from that of parasitized HbAA erythrocytes ($p < 0.001$ according to the Mann–Whitney rank sum test).

To corroborate our findings, we determined the total number of surface-presented PfEMP1 molecules per cell, using

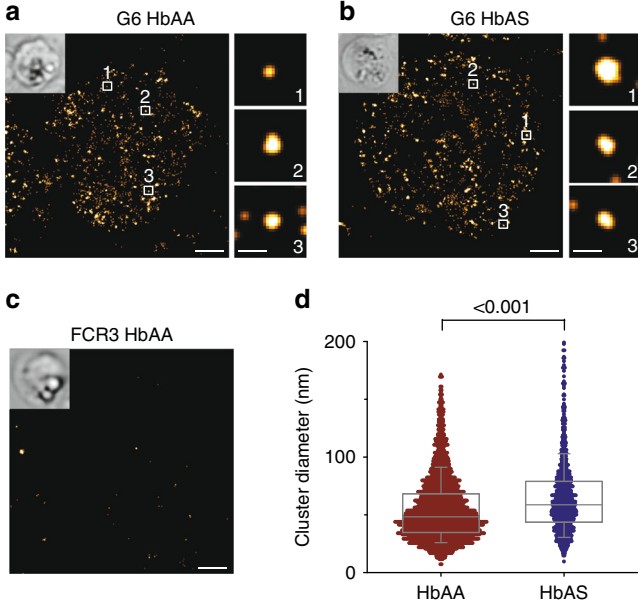

**Fig. 2** Single-molecule localization microscopy. **a**, **b** Representative PALM images of the VAR2CSA$^{mEOS2}$-expressing FCR3 mutant G6 at the trophozoite stage grown in **a** HbAA and **b** HbAS erythrocytes. PALM images were reconstructed from 12,000 imaging frames recorded with a frame rate of 10 Hz. Inserts show the corresponding bright-field images. The numbered boxes refer to magnified fluorescence clusters presented in the adjacent panel column. Bars, 1 μm in main panel and 100 nm in subpanels. **c** Representative PALM image derived from the parental strain FCR3 grown in HbAA erythrocytes. **d** Size distribution of fluorescence signal clusters from HbAA and HbAS erythrocytes infected with G6. A total of 3178 (derived from 130 cells and based on $n =$ three independent blood donors) and 999 (76 cells and $n =$ two independent blood donors) clusters, respectively, were analyzed[25]. A box plot analysis is overlaid over the individual data points, with the median and the 25 and 75% quartile ranges being shown. Statistical significance was assessed using the Mann–Whitney rank sum test

quantitative fluorescence-activated cell sorting (FACS) analysis of live erythrocytes infected with FCR3. To this end, we conjugated the monoclonal VAR2CSA antibody with an Alexa Fluor 488-labeled Fab fragment against the Fc portion. The specific fluorescence signal derived from the conjugated VAR2CSA antibody was calibrated by FACS, using microspheres coated with defined amounts of Fc-specific capture antibodies, to convert the fluorescence signal into absolute molecule numbers (supplementary Fig. 5a). As seen in Fig. 3d, the number of surface-presented PfEMP1 molecules increased with time post invasion in a sigmoidal fashion[2,3]. At the plateau level, parasitized HbAA and HbAS erythrocytes presented on average 13,000 ± 200 and 10,400 ± 500 VAR2CSA molecules per cell (Fig. 3d). A previously described FCR3 *var2csa* knock-out clone[30] and uninfected erythrocytes served as negative controls (Supplementary Fig. 5b). In parallel experiments, we determined the knob architecture by SEM and obtained knob densities of 24 ± 6 knobs μm$^{-2}$ (~3300 ± 800 knobs per cell) and 18 ± 3 knobs μm$^{-2}$ (~2200 ± 400 knobs per cell) and knob diameters of 79 ± 14 nm and 108 ± 22 nm in parasitized HbAA and HbAS erythrocytes, respectively (Fig. 3e). If one takes into consideration the average surface area of parasitized HbAA and HbAS erythrocytes at the trophozoite stage of 136 and 120 μm$^2$, respectively[31], this yields a mean number of VAR2CSA molecules per knob of 3.9 ± 0.5 (range 3.2–5.1) for infected HbAA erythrocytes and 4.7 ± 0.7 (range 3.7–5.7) for infected HbAS erythrocytes (Fig. 3f), assuming all knobs carry

PfEMP1 molecules[16]. These values are in good agreement with the values obtained by quantitative super-resolution microscopy. Note that parasitized HbAS erythrocytes have a reduced surface area owing to microvesiculation[31,32]. Comparable average numbers of PfEMP1 molecules per knob were obtained when using two other monoclonal antibodies directed against VAR2CSA in the quantitative FAC analysis (compare circles, triangles, and squares in Fig. 3f), whereas experiments using the AB01 human monoclonal antibody specific for the PfEMP1 variant IT4var32b[24] revealed no specific staining (supplementary Fig. 5c). Moreover, populations of FCR3$^{HDMEC}$ that were selected for cytoadherence to human dermal microvascular endothelial cells and which expressed predominantly *var* genes mediating interactions with ICAM-1 and/or CD36[33] did not react in our assay (supplementary Fig. 5d).

**VAR2CSA molecules are placed at the tip of a knob.** To further investigate the location of VAR2CSA on knobs, we performed immuno-electron microscopy studies in association with stereology methods, using the monoclonal antibody against VAR2CSA and a protein A gold-labeled secondary antibody to stain live parasitized erythrocytes. The labeled cells were subsequently embedded, sliced into 200 nm-thick sections and examined by transmission electron microscopy. 363 and 262 knobs from infected HbAA and HbAS erythrocytes were selected for further analysis on the basis of the sectional plane including both the base and the tip of the knob, such that the full contour of the knob was visible. Each micrographic image was then divided into sections of 8 nm each (Fig. 4a, b) and the number of gold grains per section were determined for all images (Fig. 4c). A graphical analysis of these data revealed that the probability to encounter a gold grain was the highest at the tip of the knob and declined towards the knob base (Fig. 4c). Assuming, in a first approximation, that knobs have a hemispherical shape, this allowed us to convert the data into a discrete radial distribution model and, hence, a spatial projection[34] (Fig. 4d). As seen in Fig. 4e, f, VAR2CSA molecules cluster around the top of knobs in both parasitized HbAA and HbAS erythrocytes, with the probability to encounter a PfEMP1 molecule drastically declining beyond 40 nm arc length (Fig. 4d). This cutoff value is in good agreement with the radial extension of the PfEMP1 fluorescence cluster signals, as determined by PALM.

The distribution model further allowed us to estimate the average nearest neighbor distance between VAR2CSA molecules as a function of the number of adhesins placed on a knob. For the purpose of the simulation, it was assumed that a single VAR2CSA molecule occupies a projected circular area of 110 nm$^2$ on a knob[15]. As expected, the average nearest neighbor distance decreased exponentially with increasing VAR2CSA density (supplementary Fig. 6). For three or four VAR2CSA molecules per knob, the average nearest neighbor distance between adhesins was estimated to be approximately between 18 and 14 nm, respectively.

## Discussion
Our finding of a very-low VAR2CSA molecule number per knob was unexpected. However, it is consistent with the reported inefficient trafficking and sorting of PfEMP1 molecules to the erythrocyte surface, with the vast majority of the PfEMP1 molecules remaining associated with membranous structures within the host cell compartment[35]. We considered the possibility that tagging VAR2CSA with mEOS2 might have affected trafficking efficiency of the fusion protein to the surface or that the PALM images might be compromised by signals from PfEMP1 proteins associated with intracellular membranes, particularly, if

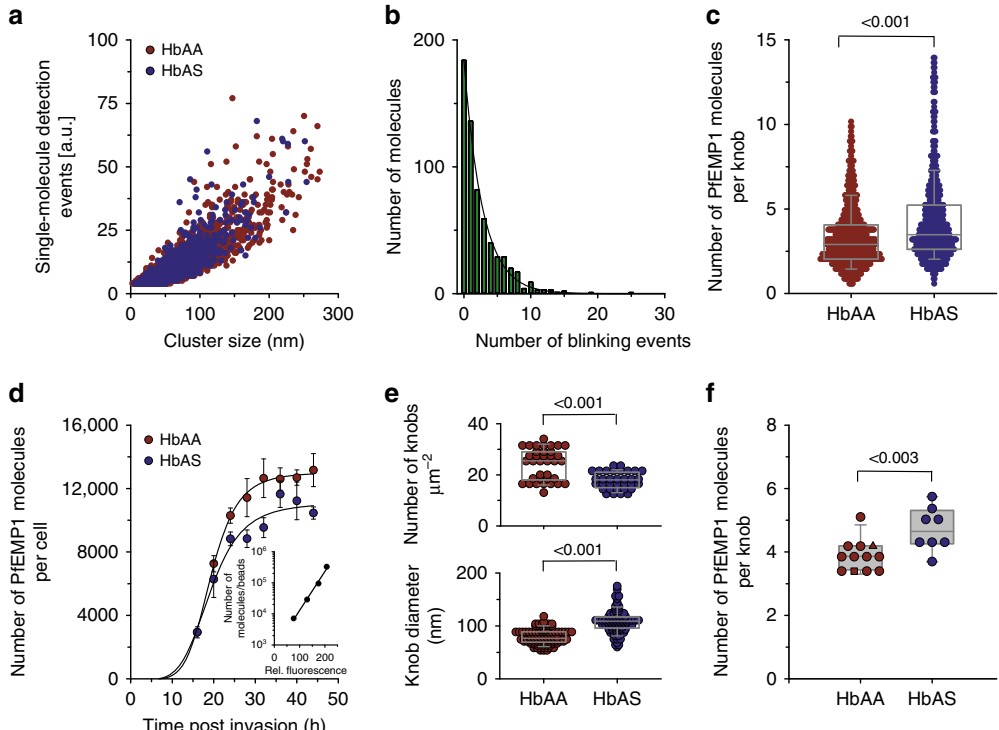

**Fig. 3** A quantitative framework for assessing the number of PfEMP1 molecules per knob. **a** Single-molecule detection events are shown as a function of the fluorescence cluster size for the G6 strain grown in HbAA (dark red dots) and HbAS (dark blue dots) erythrocytes[25]. **b** Calibration curve showing the blinking distribution curve of single mEOS2 molecules immobilized on a glass surface[25]. A single exponential decay function was fitted to the data points. **c** Number of PfEMP1 molecules per knob for G6 at the trophozoite stage grown in HbAA and HbAS erythrocytes, as determined by PALM. A total of 771 and 638 knobs, respectively, were analyzed[25]. A box plot analysis is overlaid over the individual data points, with the median and the 25 and 75% quartile ranges being shown (also in the following subfigures). Statistical significance was assessed using the Mann–Whitney rank sum test (also in the following subfigure). **d** Absolute number of surface-presented PfEMP1 molecules (VAR2CSA variant) per single erythrocyte infected with the parental FCR3[CSA] strain as a function the time post invasion, as determined by quantitative FACS analysis using the Zenon-labeled monoclonal antibody PAM 8.1 directed against VAR2CSA. The mean ± SEM are shown for n = 4 or more independent biological samples / blood donors[25]. Red and blue circles, FCR3[CSA] grown in HbAA and HbAS erythrocytes respectively. The insert shows a representative calibration curve to convert fluorescence signals into absolute molecule numbers. **e** Knob density and knob diameter for HbAA and HbAS erythrocytes infected with FCR3[CSA] (trophozoites), as determined by scanning electron microscopy[25]. Data are derived from x = three independent biological samples / blood donors. **f** Number of PfEMP1 molecules per knob for FCR3[CSA] grown in HbAA (red) and HbAS (blue) erythrocytes, as determined by quantitative FACS analysis and taking into account cell surface area and knob density[25]. Each data point corresponds to an independent biological sample based on a different blood donor. The following anti-VAR2CSA monoclonal antibodies were used: PAM 8.1 (red circles), PAM 3.1 (red triangle), PAM 4.7 (red square)

they were close to the erythrocyte plasma membrane. However, we do not think that such effects influenced the outcome of our results or the conclusions drawn from our data. First, PfEMP1 carrying transport vesicles are < 25 nm in size and, therefore, smaller than the fluorescence clusters detected in the PALM images[19]. Second, analysis of the G6 mutant expressing a genetically modified *var2csa* gene and the wild-type strain FCR3 expressing the unaltered *var2csa* gene revealed comparable numeric values. Moreover, two independent approaches, namely super-resolution microscopy and quantitative FACS analysis, gave very similar results.

A low number of PfEMP1 molecules per knob is in agreement with estimates from a previous immuno-EM study, which investigated a parasite line termed A4 that presented a PfEMP1 variant mediating cytoadhesion to ICAM-1 and CD36 (encoded by *A4var*) on the erythrocyte surface[16]. Thus, there is evidence from an independent study supporting our conclusions and suggesting that, in spite of a large sequence and functional diversity among PfEMP1 variants, our findings might be transferrable to PfEMP1 variants other than VAR2CSA, although further experiments are needed to verify this suggestion. We further acknowledge the effect the genetic background of the

parasite has on PfEMP1 presentation and knob density[36–41] and that parasites lines other than FCR3 might present a different number of VAR2CSA molecules per knob.

Immunity to clinical malaria is acquired with time and age after repeated episodes of *P. falciparum* infections and the attainment of a strain-transcending antigenic memory[42]. Accordingly, infants and young children are particularly vulnerable to malaria in endemic areas. In addition, women bear an increased risk during primigravida as parasitized erythrocytes sequestering in the intervillous space of the placenta can interfere with vital placental functions[43,44]. VAR2CSA plays a central role in pregnancy-associated malaria. It mediates cytoadhesion of parasitized erythrocytes to the chondroitin-4-sulfate (CSA) containing proteoglycan matrix that lines the intervillous space of the placenta[12,30,44]. At the same time, VAR2CSA is a target of the humoral immune response that protects women from maternal malaria during subsequent pregnancies[43,44]. It is plausible that the knob/VAR2CSA architecture is optimized for effective cytoadhesion to CSA under placental flow conditions and to restrict the antibody-mediated effector mechanisms that target it. Displaying only a few VAR2CSA molecules on the tip of knobs might meet these two challenges.

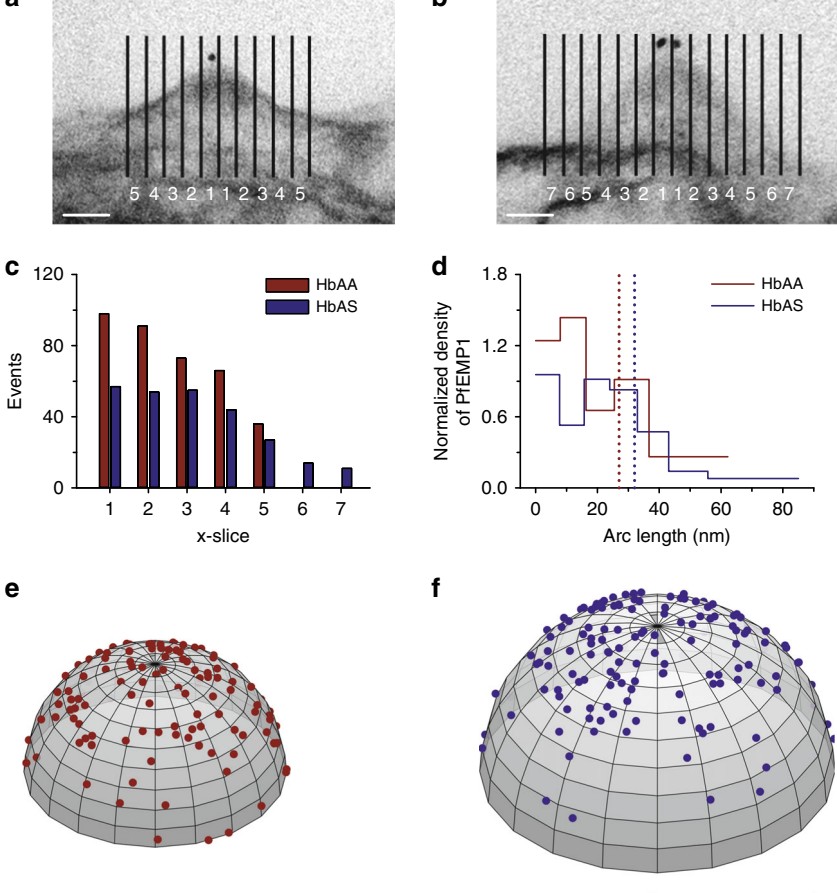

**Fig. 4** Spatial arrangement of PfEMP1 molecules on knobs. **a**, **b** Representative electron micrographic images of knobs from HbAA **a** and HbAS **b** erythrocytes infected with FCR3[CSA], labelled with the VAR2CSA specific monoclonal antibody PAM 8.1 as a primary antibody and, as a secondary antibody, a goat anti human antibody coupled to 10 nm protein A gold. For stereological analysis, the micrographic images were divided in sections of 8 nm each and the sections were numbered from the center of the knobs outward. Bar, 20 nm. **c** Number of gold grains per section[25]. **d** Distribution of PfEMP1 molecules on knobs as a function of the arc length (from center to periphery), assuming knobs have a hemispherical shape[34]. The dotted lines indicate the position of the average radial extension of the VAR2CSA[mEOS2]-derived fluorescence signal, as reference points. **e**, **f** A spatial probability distribution model showing the predicted location of 120 and 150 PfEMP1 molecules on an idealized hemispherical knob from infected HbAA **e** and HbAS **f** erythrocytes, respectively[34]. The model takes into account the different knob size in infected HbAA and HbAS erythrocytes. Bar, 50 nm

Joergensen et al. (2010) have recently shown that high avidity antibodies are required for strong binding to VAR2CSA[15]. By thinly spreading only three or four VAR2CSA molecules over the top of a knob, the chance of an antibody might be reduced to cross-link two neighboring VAR2CSA adhesins, which, in turn, increases the antibody off-rate and limits the potential impact antibody binding has on VAR2CSA functionality[15]. Modelling the distribution of three or four VAR2CSA molecules on knobs provided initial support for such a model, by suggesting an average nearest neighbor distance of 18 and 14 nm, respectively, between VAR2CSA molecules (supplementary Fig. 6), which would just be beyond the reach of IgG-class antibodies (reach: 10–15 nm)[45]. Thus, a low number of VAR2CSA molecules per knob might have evolved to reduce the risk of an antibody cross-linking and thereby incapacitating two VAR2CSA molecules.

Positioning VAR2CSA and possibly other PfEMP1 molecules at the tips of the knobs would have the immediate benefit of elevating the binding sites above the cell body, thereby increasing the probability of substrate binding[46]. By sticking out into the fluid, PfEMP1 molecules can explore a larger region of the surrounding space and are less restricted by the slow hydrodynamics close to the cell body. A similar strategy to increase the capture radius is used by leukocytes in that they harbor selectin receptors at the tips of microvilli[6]. Another positive effect of clustering receptors at the tip would be load sharing between neighboring receptors and increased probability of rebinding, a strategy which is also at play in cell-matrix and cell-cell contacts[47]. Although parasitized HbAS erythrocytes carrier more PfEMP1 molecules per knob than do HbAA erythrocytes, these adhesins seem to be aberrantly presented[17,48], which, together with the abnormal knob morphology and low knob density, would explain the impaired cytoadhesion phenotype[18]. Given that PfEMP1 interacts with the knob-associated histidine-rich protein and components of the membrane skeleton, including actin and spectrin[49,50], the aberrant display of PfEMP1 might result from distorted protein/protein interactions brought about by the elevated level of oxidative stress inherent to erythrocytes containing haemoglobin S[21]. In summary, our data revealed that knobs carry 3.3 ± 1.7 and 4.3 ± 2.5 VAR2CSA molecules placed at the tip of the knob in HbAA and HbAS erythrocytes infected with the *P. falciparum* strain FCR3. Thus, relatively few valences need to be targeted by anti-adhesion drugs or vaccines in order to prevent the disease-causing cytoadhesive behavior of infected erythrocytes.

## Methods

**Parasite culture.** The *P. falciparum* strain FCR3 and the mutant G6 were cultured, as described[51], using fresh HbAA and HbAS erythrocytes resuspended in Rosewell Park Memorial Institute 1640 medium supplemented with 5% GlutaMAX (ThermoFisher Scientific) and 5% human serum. Erythrocytes were used for infection with *P. falciparum* immediately or within 10 days after blood donation. The cultures were maintained at a haematocrit of 4%, parasitemia of less than 5% and at 37 °C under atmospheric conditions of 5% $O_2$, 3% $CO_2$, and 96% humidity. Cultures were tightly synchronized using a combination of 5% sorbitol/heparin[52].

**Generation of the *var2csa/mEOS2* mutants.** Oligonucleotide primers used in this study are listed in supplementary Table 1. Homology regions (HR) flanking the desired integration sites for mEOS2 were amplified by PCR and cloned into the pL6-BsgI vector[53]. The HR5′ and HR3′ were cloned, using *Sac*II and *Bss*HII and *Spe*I and *Afl*II, respectively. Two copies of codon optimized mEOS2 joined by a five glycine linker were synthesized by GeneArt (ThermoFisher Scientific). The fragment was subsequently cloned by in-fusion between the HRs (Clontech). The guide RNAs were cloned into the BtgZI site of the pL6-BsgI vector. The resulting vector was verified by sequencing analysis. Prior to transfection, the *P. falciparum* strain FCR3 was repeatedly panned on CSA (10 mg ml$^{-1}$) coated plastic dishes to enrich for a var2csa-expressing parasite population[30]. Transfections were performed under standard conditions, using 75 µg each of the plasmid and the Cas9-expressing vector[53]. Transfected parasites were selected on 1.5 µM DMSI and 5 nM of WR99210. Integration processes were verified by PCR of genomic DNA and sequencing analysis. Clones were obtained by limiting dilution and integration processes were again confirmed by PCR and sequencing of the resulting PCR fragments. In the case of the G6 mutant, expression of the mEOS2-tagged *var2csa* gene was confirmed by RT-PCR analysis. Complementary DNAs were generated using SuperScrip III first-strand Kit (ThermoFisher Scientific) after isolation of total mRNA using the TRIzol reagent (ThermoFisher Scientific).

**Single-molecule localization microscopy.** LabTek chambers (ThermoFisher Scientific) were covered with 0.1 mg ml$^{-1}$ concanavalin A for 60 min, washed with water and phosphate-buffered saline (PBS) before magnet purified infected erythrocytes were allow to settle on them for 10 min. After washing with PBS, cells were fixed with 4% paraformaldehyde in PBS for 10 min. Paraformaldehyde was removed and cells were kept in PBS at 4 °C until imaging. All buffers were sterile filtrated, using 0.45 µm membrane filters. Samples were imaged within 48 h after preparation. Single-molecule localization microscopy was performed on a home-built microscope operated in TIRF illumination mode[27,54]. In brief, an inverted microscope (Olympus IX71) equipped with two laser sources (LBX-405–50-CSB-PP, Oxxius,; Sapphire 568 LP, Coherent) and a × 100 oil immersion objective (PLAPO × 100 TIRFM, NA ≥ 1.45, Olympus) was used. mEOS2 was photoconverted by illumination with 405 nm (0–8 mW cm$^{-2}$), excited with 568 nm (0.26 kW cm$^{-2}$), and the fluorescence signal was filtered with a bandpass (Bright-Line HC 590/20, AHF). Single-molecule fluorescence movies (12,000 frames) were recorded with an EMCCD camera (iXon Ultra X-10971, Andor), using an integration time of 100 ms, a pre-amplifier gain of 1 and an electron multiplying gain of 200. PALM images of PfEMP1 tagged with mEOS2 were reconstructed using rapidSTORM software[55] and post-processed with the LocAlization Microscopy Analyzer (LAMA)[56]. A threshold of 63 photons for quantitative analysis of the mEOS2 signal was applied. All fluorophores were localized with a localization precision of < 30 nm. Signals from mEOS2 that appeared in consecutive camera frames within a radius of 90 nm were grouped as a single localization applying a Kalman filtering algorithm implemented in rapidSTORM. For the cluster analysis the number of blinking events from single mEOS2 clusters was determined. Mean values of cluster radii and PfEMP1 cluster per µm$^2$ were determined by DBSCAN analysis[57] implemented in LAMA, using an observation threshold of 30 nm and a minimal cluster size of 5 as threshold values. To quantify the PALM signals, the average number of detection events was determined for single mEOS2 molecules attached to a poly-lysine surface. Previous studies have shown that the blinking parameters of poly-lysine bound mEOS2 molecules is comparable to those seen for mEOS2 fusion proteins in cells[27,28,58,59], which is explained by the barrel structure of the fluorescent proteins protecting the chromophore from the environment.

**Labeling antibodies with Zenon.** The anti-VAR2CSA monoclonal antibodies PAM 8.1, PAM 4.7, and PAM 3.1[24] were conjugated with an Alexa Fluor 488-labeled Fab fragment against the Fc portion, using the Zenon IgG Labeling Kit as recommended by the supplier (ThermoFisher Scientific)[60]. Zenon labeling allowed a defined number of Alexa Fluor dyes to be attached to the monoclonal antibodies without interfering with the VAR2CSA-recognizing Fab fragment.

**Immunofluorescence.** Live *P. falciparum*-infected erythrocytes at the trophozoite stage were resuspended in PBS containing 3% bovine serum albumin (BSA) and the Zenon-labeled anti-VAR2CSA monoclonal antibody PAM 8.1 (2 µg ml$^{-1}$)[24]. Samples were incubated for 30 min at room temperature while mixing the cells suspension every 10 min. Cells were subsequently washed with PBS containing 3% BSA followed by a final wash with PBS. Images were taken with a Leica TCS SP8 confocal laser scanning microscope (excitation: 488 nm, emission: 494–530 mm,

objective: APO CS2 63 × /N.A. 1.4 oil) and deconvoluted using the Imaris software. Images were processed and analyzed using the Autoquant software.

**Quantitative FACS analysis.** The *P. falciparum* strain FCR3 was repeatedly panned over CSA prior to the analysis to enrich for a parasite population that predominately expressed *var2csa*, yielding *var2csa* expression levels of ~ 90% (in reference to all expressed *var* genes) in the population[12]. Infected erythrocytes were enriched by magnet purification from a highly synchronized parasite culture. Live cells were labeled using the Zenon-labeled anti-VAR2CSA monoclonal antibody PAM 8.1 (2 µg ml$^{-1}$). Where indicated PAM 4.7 and PAM 3.1, both recognizing the DBL5-ε domain[24], were used. The specific fluorescence signals were quantified by FACS analysis using the FACScalibur (Becton Dickinson). The fluorescence signal was subsequently calibrated using Quantum™ Simply Cellular (Bangs Laboratories, Inc) as recommended by the supplier[61]. In brief, microspheres coated with defined amounts of Fc-specific capture antibodies were incubated with 2 µg ml$^{-1}$ of the Zenon-conjugated VAR2CSA antibodies in PBS supplemented with 3% BSA for 30 min at room temperature, while mixing every 10 min. The samples were washed two times with PBS containing 3% BSA followed by a final wash with PBS. The fluorescence signals were then determined by FACS and the numbers of bound antibody determinate using QuickCal (Bangs Laboratories, Inc)[62]. Note that the same Zenon-labeled monoclonal antibody batches were used for the cell surface labeling experiment and for quantification. The FCR3 *var2csa* knock-out clone[30] and uninfected erythrocytes were analyzed in parallel. The AB01 human monoclonal antibody[24] was used as a negative staining control in parallel assays.

**SEM.** SEM of parasitized erythrocytes was conducted as described[21]. In brief, magnet purified *P. falciparum*-infected erythrocytes at the late trophozoite stage were washed with PBS and fixed with 1% glutaraldehyde[63], and stored at 4 °C. Fixed samples were deposited on poly-lysine (aqueous dilution 1:1000) coated coverslips, passed through a dehydration series in acetone. Samples were dehydrated in acetone using a critical point drying device (Leica EM CPD300). Samples were coated with a 10 nm layer of gold/palladium alloy (Leica EM ACE600), and analyzed using a Leo Gemini 1530 SEM. Images were recorded at a nominal magnification of × 10k.

**Immuno-electron microscopy.** A cell suspension of magnet purified, live *P. falciparum*-infected erythrocytes at the late trophozoite stage were washed with PBS (each step of washing and rinsing by spinning at 2100 g for 2 min, and removal of supernatant), incubated in blocking buffer (1.5% BSA and 0.1% fish skin gelatine in PBS) for 30 min, and then incubated in blocking buffer containing the anti-VAR2CSA monoclonal antibody PAM 8.1 (2 µg ml$^{-1}$) for 90 min at room temperature under slight agitation. The cells were subsequently rinsed with three changes of blocking buffer and incubated in a 1:20 dilution (in blocking buffer) of secondary antibodies (goat anti human, coupled to 10 nm protein A gold (Sigma)) for 30 min at room temperature and slight agitation. The cells were subsequently washed twice using PBS and fixed in 2% paraformaldehyde and 2% glutaraldehyde. The samples were prepared for transmission electron microscopy according to standard protocols by contrasting with osmium tetroxide and uranyl acetate followed by dehydration and epon-embedding[19]. In total, 200 nm thick sections were imaged at 15 kV using a JEOL-1200 electron microscope operating at an accelerating voltage of 80 kV. The height of knobs was determined from EM images by measuring the distance between the knob base and top.

**Stereological analysis.** Only images were considered for stereological analysis where the sectional plane ran through the knob base such that the contour of the knob was clearly visible. Each image was divided in 8 nm wide sections, starting from the center of the knob, and the positions of the gold grains were recorded with regard to the section it laid in. In a first approximation, knobs can be thought of as half-spheres, although the real knob will deviate from this shape in each realization. For the analysis, we assumed that the whole knob lies within the 200 nm-thick EM section. Therefore, the data we recorded from the images correspond to slices of the thickness $h$ through a half-sphere. If a sphere is cut by two parallel planes, the surface area of the region in-between does not depend on the exact position of the cut but only on the thickness of the slice. This allowed us to convert the count of labels to a density in a specified region without any geometric correction factors[34]. We then converted the obtained distribution such that the density is a function of arc length away from the sphere's top center point[34]. To do so, the areas needed to be calculated, as in each ring a number of different slices contribute to the average density. The densities in each ring can now be determined with an iterative calculation, starting from the outermost slice. To produce the illustrations of the 3D distribution in Fig. 4, we determined how many of the gold grains belong to each ring and then distributed the given number uniformly over the relevant ring area.

**Modeling nearest neighbor distances.** Knobs were considered as half-spheres with a radius of 40 nm. We further assumed that a single VAR2CSA molecule occupies an area of 110 nm$^2$ on a given knob, according to a previous study[15]. A predefined number of VAR2CSA molecules (e.g. 3, 5, 7, or 9), respectively circular discs with an area of 110 nm$^2$, were then placed on the surface of the half-sphere by

applying the calculated radial discrete distribution model (see above and Fig. 4d). The distance to the nearest neighbor was then calculated for each VAR2CSA molecule. The simulations included volume effects, such that the discs were not allowed to overlap. Each placement of a given number of VAR2CSA molecules was repeated more than 20,000 times to obtain 181,440 nearest neighbor distances.

**Statistics and reproducibility**. Data are given as mean ± standard deviation throughout this study, if not indicated otherwise. The number of independent biological replicates, as defined by the number of different blood donors, is indicated in the main text and/or the figure legends. Statistical analyses were performed, using the Sigma Plot (v.13, Systat) software. Statistical significance was determined using the Mann–Whitney rank sum test and the Kruskal–Wallis one way analysis of variance on ranks test, where indicated.

**Ethics approval**. The study was approved by the ethical review boards of Heidelberg University and Mannheim University. Written informed consent was given by all blood donors.

**Reporting summary**. Further information on research design is available in the Nature Research Reporting Summary linked to this article.

## Data availability
The authors declare that the data supporting the findings of this study are available within the article and its supplementary information files, or are available from the authors upon request. The original data underlying this article are available at DRYAD[64] https://doi.org/10.5061/dryad.39jf355.

## Code availability
The computer codes developed for this study are available at the following URLs: https://github.com/usschwarz/PfEMP1-Distribution or https://doi.org/10.5281/zenodo.2633095 (ref. [65]).

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

## Acknowledgements

This work was funded by the Deutsche Forschungsgemeinschaft (DFG, German Research Foundation)—Projektnummer 240245660—SFB 1129 (M.L. and U.S.S.) and SFB 807 (C.K. and M.H.). We thank Vibor Laketa for help with confocal microscopy, Marina Müller for excellent technical assistance, and Lars Hviid for the generous gift of the PAM antibodies. U.S.S and M.L. are members of the cluster of excellence CellNetworks.

## Author contributions

M.H., C.P.S., and M.L. designed the study. C.P.S., C.K., R.S., M.C. performed the experiments. J.J. did the stereological analysis. B.B. collected the HbAS blood. C.P.S., C.K., R.S., M.C., M.H., and M.L. analyzed the data. M.L. wrote the manuscript with help from C.P.S., C.K., M.C. J.J., U.S.S. and M.H. All authors participated in discussion and manuscript editing.

## Additional information

**Competing interests:** The authors declare no competing interests.

