## [Peer Review File · Communications Biology]

Editorial Note: This manuscript has been previously reviewed at another journal. This document only contains reviewer comments and rebuttal letters for versions considered at *Communications Biology*.

REVIEWERS' COMMENTS:

Reviewer #1 (Remarks to the Author):

I am satisfied with the changes made to the manuscript and the authors' responses. I strongly support publication at this stage.

Reviewer #2 (Remarks to the Author):

The authors have fully answered my comments and provided all the requested information. This is a very interesting and convincing study of interest for the malaria field and the wider field.

I have no further request.

Benoit Gamain

Reviewer #3 (Remarks to the Author):

In the revised version of their manuscript, the authors clarified several of the points I raised in my original review. They otherwise provided helpful additional information and substantially improved their publication. I would, therefore, endorse publication in *Communications Biology*.

Responses to reviewers' comments on manuscript COMMSBIO-19-0008-A

We thank the reviewers for their support and for recommending publication of our article.

Reviewer #1:

I am satisfied with the changes made to the manuscript and the authors' responses. I strongly support publication at this stage.

Reviewer #2:

The authors have fully answered my comments and provided all the requested information. This is a very interesting and convincing study of

interest for the malaria field and the wider field. I have no further request.

Benoit Gamain

Reviewer #3:

In the revised version of their manuscript, the authors clarified several of the points I raised in my original review. They otherwise provided

helpful additional information and substantially improved their publication. I would, therefore, endorse publication in Communication Biology.